# Impact of Body Mass Index on Activities of Daily Living in Patients with Idiopathic Interstitial Pneumonias

**DOI:** 10.3390/healthcare8040385

**Published:** 2020-10-05

**Authors:** Kengo Shirado, Hidetaka Wakabayashi, Keisuke Maeda, Ryo Momosaki

**Affiliations:** 1Department of Rehabilitation, Iizuka Hospital, 3-83 Yoshiomachi, Iizuka, Fukuoka 820-8505, Japan; 2Department of Rehabilitation Medicine, Tokyo Women’s Medical University Hospital, 8-1, Kawada-cho, Shinjuku-ku, Tokyo 162-8666, Japan; noventurenoglory@gmail.com; 3Department of Geriatric Medicine, National Center for Geriatrics and Gerontology, 7-430 Morioka-cho, Obu City, Aichi 474-8511, Japan; kskmaeda@ncgg.go.jp; 4Department of Palliative and Supportive Medicine, Aichi Medical University, Nagakute, Aichi 480-1195, Japan; 5Department of Rehabilitation Medicine, Mie University Graduate School of Medicine, 2-174 Edobashi, Tsu, Mie 514-8507, Japan; momosakiryo@gmail.com

**Keywords:** activities of daily living, body mass index, idiopathic interstitial pneumonias, lung diseases, malnutrition, rehabilitation, thinness

## Abstract

In patients with idiopathic interstitial pneumonias, undernutrition has a profound effect on prognosis. However, whether body mass index affects the ability to perform activities of daily living as measured by the Barthel index in patients with idiopathic interstitial pneumonias remains unknown. Therefore, we examined the impact of body mass index on the activities of daily living in inpatients with idiopathic interstitial pneumonia. We used a database constructed by the Japan Medical Data Center. Data were extracted from 2774 inpatients from participating hospitals with a diagnosis of idiopathic interstitial pneumonia. Multiple regression analysis adjusted for confounding factors was performed to determine whether body mass index classification would be independently related to change in Barthel index during hospitalization. Underweight, normal weight, overweight, and obesity numbered 473 (19%), 1037 (41), 795 (31%), and 235 (9%), respectively. Multivariable analysis showed that being underweight was independently associated with a change in Barthel index during hospitalization of −2.95 (95% confidence interval −4.82 to −1.07) points lower than being normal weight. Approximately 20% of the patients with idiopathic interstitial pneumonias were underweight. Those who were underweight had decreased independence in activities of daily living during hospitalization.

## 1. Introduction

Patients with idiopathic interstitial pneumonias commonly have low body mass index (BMI), which can lead to poor outcomes. A cohort study conducted in the United States found that in patients with idiopathic pulmonary fibrosis, lower BMI was associated with shorter survival [1]. Furthermore, BMI less than 18.5 kg/m^2^ is independently associated with mortality [2]. These findings demonstrate that, in patients with idiopathic interstitial pneumonias, undernutrition has a profound effect on prognosis. There are various controversies about applying international standards for obesity to Asian populations, and World Health Organization (WHO) expert consultations have identified cut-offs for BMI in public health in Asia and the Pacific [3], but few reports have been validated using them. 

The relationship between nutritional status and idiopathic interstitial pneumonia is unclear, but in some diseases, nutritional status has an impact on physical functions. For patients undergoing rehabilitation for chronic obstructive pulmonary disease (COPD) [4] and hospital-related deconditioning [5], malnutrition has been reported to be associated with low physical function and exercise tolerance. In obese patients with COPD, both fat-free mass and maximal exercise capacity have been found to be greater than in normal-weight patients [6]. Malnutrition in the course of non-microcellular lung cancer has been demonstrated to significantly reduce psychomotor function [7]. Studies in stroke, orthopedics, and heart disease have shown that patients who are overweight or obese in the hospital are more independent in their activities of daily living at discharge [8,9,10]. In particular, the Barthel Index at discharge in patients with acute heart failure was higher in overweight (median: 100, 95% confidence interval: 80–100) and obese (median: 100, 95% confidence interval: 80–100) than in underweight (median: 80, 95% confidence interval: 90–100) patients, and body mass index was independently associated with the Barthel Index score at discharge [8]. Collectively, these findings suggest that nutritional condition is associated with the ability to perform activities of daily living, but this association has not yet been validated in idiopathic interstitial pneumonia. In particular, whether BMI affects the ability to perform activities of daily living in patients with idiopathic interstitial pneumonias remains unknown.

Therefore, the objective of this study was to examine the impact of BMI on activities of daily living as measured by the Barthel index in inpatients with idiopathic interstitial pneumonia.

## 2. Materials and Methods 

### 2.1. Study Design

This retrospective cohort study used a database of hospitals established by the Japan Medical Data Center (JMDC). The database contained data from the diagnosis procedure combination (DPC) survey data on nearly 5 million patients from 100 acute care hospitals in Japan [11]. To make this database available for research purposes, the files were linked via an anonymous identifier to a secondary database. Consecutive patients hospitalized with idiopathic interstitial pneumonia between April 2014 and December 2018 from JMDC DPC survey data were extracted and administrative billing data and some more clinical data were gathered. The database contains a unique identifier for each hospital and the following patient data: age, sex, presence or absence of rehabilitation during hospitalization, length of hospital stay, and diagnosis and comorbidities as recorded by the attending physicians using text data in Japanese and the codes of the International Statistical Classification of Diseases and Related Health Problems, 10th Revision (ICD-10). 

### 2.2. Ethical Considerations

Because this data is anonymous, the requirements for informed consent and authorization were waived. Analysis of the JMDC database was authorized by the Teikyo University Institutional Review Board. This study was based on the Declaration of Helsinki in 1964 and subsequent amendments to ethical standards.

### 2.3. Subjects and Research

We identified patients over 20 years old who were hospitalized at participating hospitals and diagnosed with idiopathic interstitial pneumonia (ICD-10, code J84). Inpatients admitted for purposes other than treatment, and missing data on death on admission, BMI, and/or Barthel Index score [12] were excluded. BMI categories at admission were assigned based on the Asian-Pacific cutoff points of underweight (<18.5 kg/m^2^), normal weight (18.5 to 22.9 kg/m^2^), overweight (23.0 to 27.4 kg/m^2^) and obesity (≥27.5 kg/m^2^) [3]. The ability to perform activities of daily living was assessed with the Barthel Index. The Barthel Index consists of 10 items (feeding, moving from wheelchair to bed and back, doing personal toilet, getting on and off the toilet, bathing self, walking on a level surface, ascending and descending stairs, dressing and undressing, continence of bowels, and controlling bladder), scored on a scale from 0 (demonstrated complete dependence) to 100 (demonstrated complete independence). A total score of 0–20 indicates “perfect” dependence, 21–60 indicates “severe” dependence, 61–90 indicates “moderate” dependence, and 91–99 indicates “mild” dependence [13]. In this study, the Barthel Index was assessed by the attending physician or nurse in charge. Comorbidities were evaluated using the updated Charlson Comorbidity Index [14] according to ICD-10 codes. The Brinkman index [15] was used as a measure of the amount of smoking. The Japan Coma Scale (JCS) score was used to evaluate the assessment of consciousness levels at admission [16,17].

### 2.4. Statistical Analysis

The statistical analysis was conducted using EZR version 1.37 software [18]. The chi-square test, Kruskal–Wallis test, and Bonferroni multiple comparisons test were used to analyze differences between BMI categories. To investigate whether BMI categorization was independently related to change in Barthel Index scores during hospitalization, multiple regression analysis was performed. Normal weight among the four BMI categories was used as a reference and entered into the multivariate analysis. Individual factors were age, sex, Barthel Index score at admission, updated Charlson Comorbidity Index score [19], emergency transport, Fletcher–Hugh–Jones classification [20], the presence or absence of oxygen therapy during hospitalization, and the presence or absence of rehabilitation during hospitalization. We considered a *p* value <0.05 to be statistically significant. This study of the present study conforms to the Strengthening the Reporting of Observational Studies in Epidemiology (STROBE) statement. 

## 3. Results

There were 4233 inpatients with idiopathic interstitial pneumonia in the JMDC DPC database over the study period. It excluded 3 patients <20 years old, 849 patients who died during hospitalization, 234 patients who were admitted for purposes other than to treat idiopathic interstitial pneumonia, and 607 patients with missing data. Thus, we analyzed 2540 patients with idiopathic interstitial pneumonias in this study. 

Table 1 shows the proportions of different idiopathic interstitial pneumonias and Table 2 shows the clinical and demographic data for all patients. 

Barthel Index score at discharge (*p* < 0.001), change in Barthel Index during hospitalization (*p* = 0.001), Fletcher–Hugh–Jones classification (*p* < 0.001) were significantly different among BMI categories. 

Table 3 shows the results of multiple regression analysis of the change in Barthel Index during hospitalization. Being underweight was independently related to a Barthel Index of −2.95 (95% confidence interval −4.82 to −1.07) points lower than being normal weight.

## 4. Discussion

We examined the impact of BMI on activities of daily living in inpatients with idiopathic interstitial pneumonias. As far as we know, this study is the first to investigate the association between BMI and activities of daily living in patients with idiopathic interstitial pneumonia. 19% of the patients were underweight. Those underweight had decreased independence in activities of daily living during hospitalization. 

19% of the patients with idiopathic interstitial pneumonia were underweight. It has been shown that patients with interstitial pneumonia in Western countries tend to be overweight or obese (mean BMI = 25.3–29.7 kg/m^2^) [1,21,22,23,24]. On the other hand, patients in Japan and Korea tend to have low-normal weight (BMI = 21.2–23.9 kg/m^2^) [2,25,26]. There has been some controversy regarding the application of international standards for obesity to Asian populations. First, there are growing indications of a higher prevalence of type 2 diabetes mellitus and increased evidence of cardiovascular risk factors in areas of Asia where the average BMI is less than the cut-off point defined as overweight by the WHO classification. Second, there are growing indications that the association between BMI, body fat percentage, and body fat distribution varied across populations. In response to these controversies, the WHO expert consultation identified further potential public health action points [3]. Furthermore, the Global Leadership Initiative on Malnutrition, a global diagnostic standard for undernutrition, has a lower BMI in the diagnosis of malnutrition in Asia than elsewhere [27]. Therefore, although BMI cutoffs are different in Asian countries including Japan compared to Western countries, in the present study, there were more underweight than obese patients, even using the Asian BMI cutoffs.

Underweight inpatients with idiopathic interstitial pneumonias had decreased independence in activities of daily living during hospitalization. The prevalence of sarcopenia in idiopathic pulmonary fibrosis at a tertiary care hospital was shown to be 24.9%, and low skeletal mass has been found to be associated with all-cause mortality [26]. Additionally, patients with interstitial lung disease frequently are clinically undernourished [28]. For idiopathic pulmonary fibrosis, depleted nutrition as assessed by BMI is a common finding and, in addition, is an independent predictor of mortality [1,2]. Malnutrition leads to changes in myosin content and muscle atrophy, altering the bio-energy of the muscles and it down-regulates energy-dependent cell membrane pumping of muscle fibers, and ultimately changes in muscle contraction and relaxation, reduced force-generating capacity, and increased fatigability [29,30,31]. Patients who are overweight or obese are also more likely to have more muscle mass than those who are underweight [32]. Furthermore, pulmonary rehabilitation for interstitial lung disease patients improves 6-min walking distance, symptoms, and health-related quality of life [33]. Therefore, in addition to pulmonary rehabilitation, nutritional assessment and improvement are important for treating malnutrition and activities of daily living in patients with idiopathic interstitial pneumonia.

This study has a few limitations. First, because it is a retrospective study, we were limited in our ability to determine the causal relation between BMI and the observed outcome. Second, the JMDC DPC database used in this study lacks detailed information on sarcopenia, skeletal mass, rehabilitation programs, and the severity of interstitial lung disease as assessed by the ILD-GAP index. In order to determine the association between BMI and the ability to perform activities of daily livings in idiopathic interstitial pneumonia further research is needed. Third, our subjects included a mix of patients with acute illnesses such as acute interstitial pneumonia and cryptogenic organizing pneumonia, as well as acute exacerbations of idiopathic pulmonary fibrosis and nonspecific interstitial pneumonia, and the type of idiopathic interstitial pneumonia could affect the results. However, nearly half of the patients in this study were of the unknown idiopathic interstitial pneumonia type, so it was difficult to distinguish them clearly. Therefore, to adjust for acute illnesses that may affect outcomes, we performed a separate statistical analysis by adding emergency transport as new covariates to the multivariate analysis looking at the impact of BMI on changes in Barthel Index score during hospitalization.

## 5. Conclusions

In this retrospective cohort study, 19% of inpatients with idiopathic interstitial pneumonias were underweight, and the underweight patients had decreased independence in activities of daily living during hospitalization. Therefore, combined pulmonary rehabilitation and nutritional improvement may be important in treating underweight patients with idiopathic interstitial pneumonias.

## Figures and Tables

**Table 1 healthcare-08-00385-t001:** Types of idiopathic interstitial pneumonias.

Type	*n* (%)
Idiopathic pulmonary fibrosis	410 (16.1)
Nonspecific interstitial pneumonia	199 (7.8)
Acute interstitial pneumonia	299 (11.8)
Cryptogenic organizing pneumonia	425 (16.7)
Desquamative interstitial pneumonia	1 (0.0)
Respiratory bronchiolitis-associated interstitial lung disease	2 (0.1)
Lymphocytic interstitial pneumonia	3 (0.1)
Details unclear	1201 (47.3)

**Table 2 healthcare-08-00385-t002:** Baseline demographic and clinical characteristics ^1^.

	Overall *N* = 2540	Underweight *N* = 473, 19%	Normal Weight *N* = 1037, 41%	Overweight *N* = 795, 31%	Obesity *N* = 235, 9%	*p*-Value
Age (years)						<0.001
20–64	388 (15.3)	56 (11.8)	133 (12.8)	125 (15.7)	74 (31.5)	
65–74	810 (31.9)	131 (27.7)	332 (32.0)	279 (35.1)	68 (28.9)	
≥75	1342 (52.8)	286 (60.5)	572 (55.2)	391 (49.2)	93 (39.6)	
Male (gender)	1647 (64.8)	253 (53.5)	662 (63.8)	576 (72.5)	156 (66.4)	<0.001
Barthel Index at admission	79.88 (31.02)	71.51 (35.25)	78.60 (31.51)	84.68 (27.45)	86.15 (26.81)	<0.001
Barthel Index at discharge	86.69 (26.43)	77.68 (32.52)	86.44 (26.49)	90.90 (21.81)	91.68 (21.87)	<0.001
Change in Barthel Index during hospitalization	6.81 (21.39)	6.17 (23.32)	7.84 (21.15)	6.22 (21.22)	5.53 (18.79)	0.232
Change in Barthel Index during hospitalization	0 (0–5)	0 (0–5)	0 (0–5)	0 (0–0)	0 (0–0)	0.143
Fletcher–Hugh–Jones						<0.001
1	426 (17.9)	63 (14.8)	169 (17.3)	91 (19.8)	103 (20.0)	
2	418 (17.6)	62 (14.6)	155 (15.9)	89 (19.4)	112 (21.7)	
3	403 (17.0)	56 (13.2)	163 (16.7)	84 (18.3)	100 (19.4)	
4	626 (26.4)	113 (26.6)	272 (27.9)	130 (28.3)	111 (21.6)	
5	501 (21.1)	131 (30.8)	216 (22.2)	65 (14.2)	89 (17.3)	
Emergency transport	304 (12.0)	72 (15.2)	139 (13.4)	76 (9.6)	17 (7.2)	0.001
JCS at admission						0.071
0	2418 (95.2)	440 (93.0)	990 (95.5)	760 (95.6)	228 (97.0)	
≥1	122 (4.8)	33 (7.0)	47 (4.5)	35 (4.4)	7 (3.0)	
Oxygen therapy	1197 (47.1)	235 (49.7)	502 (48.4)	221 (45.5)	239 (43.9)	0.197
Brinkman Index	1493.20 (3108.40)	1798.71 (3541.62)	1475.63 (3130.19)	1234.64 (2678.06)	1831.80 (3362.01)	0.005
CCI	1.22 (1.42)	1.23 (1.66)	1.20 (1.40)	1.24 (1.27)	1.30 (1.44)	0.769
Length of stay (day)	24.29 (23.66)	27.36 (25.69)	24.82 (22.68)	22.37 (24.51)	22.27 (19.81)	0.001
Rehabilitation during hospitalization	1093 (43.0)	235 (49.7)	436 (42.0)	325 (40.9)	97 (41.3)	0.013

^1^ Figures are presented as mean (standard deviation) or number of participants (%) unless otherwise indicated. JCS, Japan Coma Scale; CCI, Charlson Comorbidity Index.

**Table 3 healthcare-08-00385-t003:** Multiple regression analysis of change in Barthel Index during hospitalization.

	Coefficient	95%CI	GVIF	*p*-Value
		Lower Bound−Upper Bound		
BMI			1.08	
Underweight	−2.95	−4.82–−1.07		0.002
Normal weight	Reference			
Overweight	0.29	−1.27−1.86		0.715
Obesity	0.74	−1.69−3.16		0.551
Age			1.13	
20–64	Reference			
65–74	−1.81	−3.86−0.25		0.085
≥75	−4.32	−6.30–−2.33		<0.001
Sex	2.15	0.73−3.57	1.06	0.003
Emergency transport	1.96	−0.30−4.21	1.17	0.089
Barthel Index at admission	−0.49	−0.51–−0.46	1.41	<0.001
CCI	0.14	−0.34−0.61	1.03	0.572
Fletcher–Hugh–Jones			1.34	
1	Reference			
2	1.05	−1.16−3.26		0.351
3	1.00	−1.24−3.24		0.380
4	−0.01	−2.11−2.09		0.990
5	−4.74	−7.07–−2.41		<0.001
Oxygen therapy	−0.53	−1.97−0.92	1.20	0.476
Rehabilitation during hospitalization	−1.31	−2.69−0.07	1.07	0.062

CI, confidence interval; CCI, Charlson Comorbidity Index; JCS, Japan Coma Scale; GVIF, generalized variance inflation factor.

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
