# Peer review of "Impact of Body Mass Index on Activities of Daily Living in Patients with Idiopathic Interstitial Pneumonias"

_healthcare, 2020, doi:10.3390/healthcare8040385_

Round 1

Reviewer 1 Report

The authors examined a database constructed by the Japan Medical Data Center and demonstrated that underweight patients with IIP had smaller effects by rehabilitation on activity. It is interesting to have examined effect of rehabilitation according BMI, but I want to note some concerns.

  1. The authors used Medical data base to collect consecutive patients who hospitalized between Apr. 2014 and Dec. 2018. However, it should be mentioned what were reasons for the admissions. They all admitted for rehabilitation? Or were all kind of admissions included in the analysis? If all kind of admissions were included, the results were greatly affected by the reasons for the admission. The analysis should be done by the reason for the admission. Actually, I think the analysis should be done only in patients who admitted specifically for pulmonary rehabilitation.
  2. The authors noted in Introduction that whether BMI affects rehabilitation outcome in patients with IIP remains unknown. They also noted that the objective of this study was to examine the impact of BMI on activity of daily living in inpatients with IIP. It is quite vague what was objective of this study.
  3. The Barthel Index is usually used to assess disability. It is not adequate for evaluating the effect of rehabilitation.
  4. It should be noted who assessed the score for the Barthel Index.
  5. It is quite skeptical that 309 patients with AIP were included. Patients with AIP is quite rare and, if existed, must have been under critical situations. COP is also recognized as acute disease according to the recent guideline. On the other hand, IPF and NSIP are chronic diseases. It is not adequate that patients with IPF, AIP, and OP were simultaneously included in the study.
  6. Was the rehabilitation unified? Because this study was analysis of medical database, it is understandable that unifying the content of rehabilitation was difficult. And physical therapy, occupational therapy, and speech-language-hearing therapy are different treatment. They should not be recognized same as rehabilitation.

Author Response

Authors' Responses to Reviewer's Comments (Reviewer 1)  

Thank you for reviewing our manuscript entitled “Impact of body mass index on activities of daily living in patients with idiopathic interstitial pneumonias”.

We attempted to address each of the suggestions and criticisms raised by the reviewer.
Please see the attachment.

Reviewer 2 Report

Abstract, background and discussion

  • The main comparison, Barthel Index, is not clearly defined and connected with health function and nutrition. This needs to be indicated early in the abstract and introduction and then connected in the discussion. In fact, since BI is an indicator of active daily living skills and has cutoffs, these need to be provided.along with the explanation ie, 15-20 - significant disability.
  • BMI is a proxy measure of malnutrition in older adults, but may not be a good indicator of malnutrition in younger populations, particularly Asian populations where BMI cutoffs have been suggested by WHO to be lower than European/standard BMI cutoffs.
  • The comparison of weight classification and pneumonia outcome has not been done in the Japanese population. Since it has been done in other populations, the novelty of this research is average. Nevertheless, it has the potential to make a valuable contribution to the literature since there are conflicting viewpoints concerning BMI with age and race. This needs to be indicated in the introduction and discussion.

Methods:

  • Since there are no people at the oldest classification who are obese, I suggest regrouping that age-group (90 and over) and combining it with 75 and older.
  • The view of BMI in older Asian adults and health outcomes is mixed. BMI cutoffs for all Asians should be lower than for other races (Black/White) due to percent body fat being higher for Asians at lower BMIs. At the same time, BMIs for older adults should be higher than middle-aged and younger adults for better health outcomes. These guidelines need to be discussed and how using different cutoffs could effect the results.
  • Re-run the statistics with the cutoffs suggested for Asians from WHO for BMI and with regrouping of age-groups. You used WHO BMI for general populations (European) (lines 79-81). From 75 onward underweight is an issue for health; however, what is underweight for Asians?

Minor comments:

The methods section would be better with subheadings rather than short paragraphs (Study population, ethics, statistical analysis ...).

Consider how you use paragraphs throughout the manuscript. Each paragraph should have a topic sentence and be at least 5 sentences long. Consider combining the ideas of shorter paragraphs.

Author Response

Thank you for reviewing our manuscript entitled “Impact of body mass index on activities of daily living in patients with idiopathic interstitial pneumonias”.
We attempted to address each of the suggestions and criticisms raised by the reviewer.
Please see the attachment.

Reviewer 3 Report

the article "Impact of body mass..." presents a study well structured but limited in scope.
Please incorporate further discussion with other scientific studies that incorporate BMI as a health indicator, highlighting the quantitative differences between studies in Western countries and Asian ones.

Author Response

(The authors gave the same response as above.)

Round 2

Reviewer 1 Report

The authors revised the manuscript and it became a bit easy to be read. But I still concern about ununiformity of types of admissions and diseases. And I also think that the Barthel Index is a tool for evaluating disability but not activity.

Author Response

Thank you for reviewing our manuscript entitled “Impact of body mass index on activities of daily living in patients with idiopathic interstitial pneumonias” again.
We attempted to address each of the suggestions and criticisms raised by the reviewer.
Please see the attachment.

Reviewer 2 Report

Dear Authors,

You have addressed my comments and made the necessary revisions. Just add 'as measured by the Barthel index' in the abstract and again for the objective. I attached a copy of your manuscript with this comment near the corresponding sentences.

Author Response

Thank you for reviewing our manuscript entitled “Impact of body mass index on activities of daily living in patients with idiopathic interstitial pneumonias” again.
We attempted to address each of the suggestions and criticisms raised by the reviewer.
Please see the attachment.

This manuscript is a resubmission of an earlier submission. The following is a list of the peer review reports and author responses from that submission.